# Growth, Antioxidant Capacity, and Liver Health in Largemouth Bass (*Micropterus salmoides*) Fed Multi-Strain Yeast-Based Paraprobiotic: A Lab-to-Pilot Scale Evaluation

**DOI:** 10.3390/antiox13070792

**Published:** 2024-06-28

**Authors:** Jie Wang, Xiaoze Xie, Yangyang Liu, Jiacheng Liu, Xiaofang Liang, Hao Wang, Gang Li, Min Xue

**Affiliations:** 1National Aquafeed Safety Assessment Center, Institute of Feed Research, Chinese Academy of Agricultural Sciences, Beijing 100081, China; wangjie03@caas.cn (J.W.); xiexiaoze@caas.com (X.X.); liuyangyang@caas.cn (Y.L.); liujiacheng@caas.cn (J.L.); liangxiaofang01@caas.cn (X.L.); wanghao01@caas.cn (H.W.); 2National Fisheries Technology Extension Center, Beijing 100125, China; muzi5121@163.com

**Keywords:** largemouth bass, multi-strain yeast-based paraprobiotics, pilot-scale experiment, antioxidant capacity, immune responses, liver health

## Abstract

A multi-strain yeast-based paraprobiotic (MsYbP) comprising inactive cells and polysaccharides (β-glucan, mannan oligosaccharides, and oligosaccharides) derived from *Saccharomyces cerevisiae* and *Cyberlindnera jadinii* could ensure optimal growth and health in farmed fish. This study assessed the impact of an MsYbP on the growth, immune responses, antioxidant capacities, and liver health of largemouth bass (*Micropterus salmoides*) through lab-scale (65 days) and pilot-scale (15 weeks) experiments. Two groups of fish were monitored: one fed a control diet without the MsYbP and another fed 0.08% and 0.1% MsYbP in the lab-scale and pilot-scale studies, respectively (referred to as YANG). In the lab-scale study, four replicates were conducted, with 20 fish per replicate (average initial body weight = 31.0 ± 0.8 g), while the pilot-scale study involved three replicates with approximately 1500 fish per replicate (average initial body weight = 80.0 ± 2.2 g). The results indicate that the MsYbP-fed fish exhibited a significant increase in growth in both studies (*p* < 0.05). Additionally, the dietary MsYbP led to a noteworthy reduction in the liver function parameters (*p* < 0.05), such as alanine aminotransferase (ALT), aspartate aminotransferase (AST) and alkaline phosphatase (AKP), and hepatic nuclear density, indicating improved liver health. Furthermore, the dietary MsYbP elevated the antioxidative capacity of the fish by reducing their malondialdehyde levels and increasing their levels and gene expressions related to antioxidative markers, such as total antioxidant ca-pacity (T-AOC), total superoxide dismutase (T-SOD), glutathione peroxidase (GSH-Px), catalase (CAT), nuclear factor erythroid 2-related factor 2 (*nrf2*) and kelch-1ike ech-associated protein (*keap1*) in both studies (*p* < 0.05). In terms of hepatic immune responses, the lab-scale study showed an increase in inflammation-related gene expressions, such as interleukin-1β *(il-1β*) and transforming growth factor β1 (*tgf-β1*), while the pilot-scale study significantly suppressed the expressions of genes related to inflammatory responses, such as tumor necrosis factor α (*tnfα*) and interleukin-10 (*il-10*) (*p* < 0.05). In summary, our findings underscore the role of dietary multi-strain yeast-based paraprobiotics in enhancing the growth and liver health of largemouth bass, potentially through increased antioxidative capacity and the modulation of immune responses, emphasizing the significance of employing yeast-based paraprobiotics in commercial conditions.

## 1. Introduction

Global aquaculture, a pivotal contributor to food protein production, has undergone substantial growth in recent decades. In 2020, it constituted 49.2% of the market consumption of aquatic animals [1]. However, the adoption of high stocking densities and diverse formulated diets in modern aquafarming frequently results in unpredictable diseases, exerting adverse effects on host growth and liver health [2]. To optimize growth and liver health in farmed fish, thereby mitigating production losses and augmenting farm revenue, the incorporation of various feed additives has become an integral aspect of existing nutritional regulations and health promotion strategies in aquafeed [3,4,5].

Paraprobiotic feed additives consist of inactivated probiotics or their cell extraction products, offering beneficial effects on host health when administered in appropriate amounts [6]. Yeast, a promising and sustainable ingredient, harbors various bioactive substances, including antimicrobial peptides, β-glucans, mannan-oligosaccharides, and nucleotides, rendering it an ideal source for paraprobiotics [7]. Gyan et al. (2019) reported that yeast-derived antimicrobial peptides effectively inhibit the overgrowth of intestinal pathogenic bacteria, thereby enhancing disease resistance in aquatic animals [8]. Furthermore, yeast-based β-glucans, which act as natural immune-regulating substances, stimulate the immune response by interacting with immune receptors, for example, toll-like receptors and C-type lectin families [9,10,11,12,13], and enhancing antioxidant responses in fish [14]. Yeast-derived mannan oligosaccharides (MOS) have been widely observed to modulate intestinal microbiota compositions and inhibit the adherence of pathogenic bacteria, benefiting the hosts’ health [15,16]. Generally, dietary yeast-based paraprobiotics have been shown to have beneficial effects on the growth and health of various fish species, including, but not limited to, tilapia (*Oreochromis mossambicus*) [14], rainbow trout (*Oncorhynchus mykiss*) [17], hybrid sturgeon (*Acipenser baerii*♂ × *A. schrenckii*♀) [18], common carp (*Cyprinus carpio*) [19], grass carp (*Ctenopharyngodon Idella*) [20], European seabass (*Dicentrachus labrax*) [21], and largemouth bass [22].

In our study, the multi-strain yeast-based paraprobiotic (MsYbP) comprised inactive cells and polysaccharides (e.g., β-glucan, mannan oligosaccharides, and oligosaccharides) derived from *Saccharomyces cerevisiae* and *Cyberlindnera jadinii*. Compared to a single-yeast product (*S. cerevisiae*), this mixture has been reported to activate more immuno-receptors in European seabass [21]. Additionally, rainbow trout’s intestinal mucosal response is positively impacted by MsYbP supplementation through its interaction with two toll-like receptor signaling pathways. Also, it improved antigen transport to the basal mucosal-associated lymphocytes [23]. In our recent study, we discovered that MsYbP supplementation could enhance the growth performance and improve the gut health of largemouth bass fed a diet with 23.5% cottonseed protein concentrate [22]. Moreover, it mitigated the adverse impacts of a cottonseed protein concentrate-based diet (42%) on liver health in largemouth bass through the regulation of the gut microbiota and bile acid crosstalk [24]. However, these relevant studies were primarily small-scale and short-term. There have been no large-scale, long-term pilot studies, and there has been limited research conducted on antioxidants and liver health so far.

Largemouth bass, being an economically significant carnivorous fish in North America and China, have traditionally been primarily fed with forage fish rather than formulated diets. However, in 2019, there was a significant shift in the feeding pattern, with formulated diets accounting for 80% of largemouth bass aquaculture [25]. Compared to forage fish, formulated diets for largemouth bass have posed challenges, such as the risk of inducing metabolic liver disease [26], as largemouth bass have a low tolerance for the carbohydrates in formulated diets [27,28]. To address this issue, supplementation with yeast-based paraprobiotics in formulated diets has been widely used to enhance their efficacy and alleviate the potential adverse impacts on the liver health of largemouth bass [29].

Hence, our study aims to assess the effects of dietary multi-strain yeast-based paraprobiotics on the growth performance, immune responses, and antioxidant capacities of largemouth bass in lab-scale and pilot-scale studies and to determine whether yeast-based paraprobiotics could improve the liver health of largemouth bass fed a formulated diet.

## 2. Materials and Methods

### 2.1. Diets

Two isoenergetic (22 MJ/kg gross energy) and isonitrogenous (51% crude protein) meals were developed for the lab-scale study: a control diet and the control diet supplemented with 0.08% MsYbP (referred to as YANG), as shown in Table 1. In the pilot-scale study, the control diet was purchased from a feed company (46% crude protein, 10% crude lipid, and 16% ash), and the YANG diet was formulated based on the control diet supplemented with 0.1 % MsYbP via the coating process.

### 2.2. Experimental Conditions and Fish

Feeding trials were conducted following the Laboratory Animal Welfare Guidelines of China (Ministry of Science and Technology, Decree No.2, issued in 2021). Largemouth bass were obtained from the Tianjin Yuqing Aquatic Co., Ltd. (Tianjin, China) and given a commercial diet for 14 days prior to the feeding experiments in order to allow their systems to adjust.

The lab-scale study was performed in an indoor recirculation aquaculture system with eighty tanks (vol. = 256 L) at the Nankou Base of the Institute of Feed Research, Beijing, China. The tanks were equipped with a mechanic- and bio-filter system, semi-automatic water-exchanged system, and lamp system (400 lx intensity; 12 h light/12 h dark). Throughout the trial, the water’s temperature stayed constant at 24 °C. Its dissolved oxygen saturation level stayed above 8 mg/L, while its pH varied between 7.5 and 8.5. The fish were arranged in 4 tanks/diet (20 fish/tank; mean initial body weight = 31.0 ± 0.8 g) at random and fed to satiation twice daily for 65 days.

The pilot-scale study was performed in a vertical aqua-farming system with six containers (vol. = 42,772 L, length = 6.1 m, width = 2.4 m, depth = 2.9 m) at the National Fisheries Technology extension center, Beijing, China. These containers were equipped with a mechanic- and bio-filter system, semi-automatic water-exchanged system, and natural light system. During the trial, the water’s pH fluctuated from 8 to 8.5, its dissolved oxygen saturation level remained steady 8 mg/L, and its temperature ranged from 27 to 29 °C. The fish were randomly distributed among the containers (3 containers/diet, with about 1500 fish/container; mean initial body weight = 80.0 ± 2.2 g) and fed to satiation twice daily for 15 weeks.

### 2.3. Sampling

At the beginning of the experiment, the fish were weighed to measure their initial body weight. Before the sampling, the fish were starved for one day and sampled according to our previous description [22]. Briefly, 16 fish per dietary treatment in the lab-scale study (4 fish/tank) and 12 fish per dietary treatment in the pilot-scale study (4 fish/container), respectively, were sampled, and their body weight and length were measured, along with the weights of their viscera, liver, and visceral adipose. Blood samples (8 fish per diet) were collected using a disposable syringe, and the plasma was obtained by centrifugation at 4 °C for 10 min. The liver samples were taken close to the bile duct (12 fish/diet; around 0.12 mm^3^ square), and then stored in a 4% paraformaldehyde solution for histological observation. More liver samples were collected from 8 fish per diet, frozen in liquid nitrogen, and then stored in −80 °C for analysis of the liver lipids and function.

### 2.4. Hepatic Histology and Immunofluorescence

Regarding the hepatic histology, the liver samples were stained with hematoxylin and eosin (H&E) according to standard protocols. Briefly, after soaking in a 4 % paraformaldehyde solution for 24 h, the liver samples were cut into 6 µm pieces and preserved in paraffin. The slices were then photographed using TissueFAXS Imaging software 7.0 after being stained with hematoxylin–eosin and Sirius Red.

The liver sections’ immunofluorescence was determined according to our previous description [25]. The antibodies of caspase-3 (Gene Tex, GTX54180, dilution 1:500) (Irvine, CA, USA) and activated caspase-3 (Abcam, ab13847, dilution 1:300) (Cambridge, UK) were used. The fluorescence signals were captured using a confocal microscope (Zeiss LSM700, Oberkochen, Germany) in merge format.

### 2.5. Hematological Assays and Antioxidative Activity Assays

The plasma alanine aminotransferase (ALT), aspartate aminotransferase (AST), and alkaline phosphatase (AKP) were detected using reagent kits following their protocols (Nanjing Jiancheng Bioengineering Institute, Nanjing, Jiangsu, China). The activities of the antioxidative parameters, including the malondialdehyde (MDA), total antioxidant capacity (T-AOC), total superoxide dismutase (T-SOD), glutathione peroxidase (GSH-Px), and catalase (CAT), were detected by commercial reagent kits (Nanjing Jiancheng Bioengineering Institute, Nanjing, Jiangsu, China). The plasma reactive oxygen species (ROS) and liver inflammation factors, including tumor necrosis factor α (*tnfα*), interleukin-1β (*il-1β*), and interleukin-10 (*il-10*), were measured using ELISA kits (Meimian Industrial Co., Ltd.; Suzhou, Jiangsu, China).

### 2.6. Quantitative Real-Time PCR

RNA extraction, cDNA synthesis, and real-time PCR were carried out according to the previous descriptions by Yu et al. (2019) [25]. All primer sequences and amplification efficiencies of the genes are shown in Table 2. The template volume was adjusted for normalization using the endogenous reference, the ef-1α gene (GenBank accession no. KT827794). The real-time PCR data were calculated via the 2^−∆∆Ct^ method [30].

### 2.7. Statistical Analysis

The normality and homogeneity of variance were assessed using the Shapiro–Wilk test and the standard QQ plot, respectively. If the distribution of the data was not normal, it was transformed to achieve a normal distribution. For statistical comparisons, T-tests were used to compare the normally distributed variables between the control and YANG treatments using the SPSS 22.0. All data were presented as the means ± standard error of the mean (SEM), and *p* values less than 0.05 were considered significant differences. The figures were generated by GraphPad Prism 9.0 (GraphPad Software Inc., San Diego, CA, USA).

## 3. Results

### 3.1. Growth Performance and Morphometrics

In both the lab-scale and pilot-scale studies, the largemouth bass fed the diet containing the multi-strain yeast-based paraprobiotics (YANG group) exhibited a significant increase in their final body weight (FBW) and weight gain rate (WGR) compared to the control group (Table 3, *p* < 0.05). In the lab-scale study, there was no significant difference in the survival rate between the control group and the YANG group (Table 3, *p* > 0.05). However, the YANG-fed fish showed a significantly higher feed consumption than the control-fed fish (Table 3, *p* < 0.05). In the pilot-scale study, the data on survival rate and feed consumption were not available because they are the intellectual property of a commercial enterprise.

The morphometric parameters, including the condition factor (CF), viscero-somatic index (VSI), and hepatosomatic index (HSI), were not significantly influenced by the dietary treatments in either study (Table 3, *p* > 0.05). However, the visceral adipose index (VAI) was significantly increased in the fish fed the YANG diet compared to the control diet in both studies (Table 3, *p* < 0.05).

### 3.2. Liver Histological Examination

Two hepatic phenotypes were observed in our studies, as illustrated in Figure 1A. In the phenotype with no obvious abnormalities, the hepatocytes displayed intact morphological shapes, and the microvasculature was evenly distributed in the extracellular space (Figure 1A). In contrast, the nuclear dense phenotype exhibited shrunken hepatocellular shapes with tightly packed nuclei and an expanded microvasculature in the extracellular space (Figure 1A). The immunofluorescence staining results revealed that liver tissues with no obvious abnormalities showed a high expression of caspase 3 but low cleaved caspase 3 expression, whereas the nuclear-dense phenotype exhibited a low expression of caspase 3 but high cleaved caspase 3 expression (Figure 1A).

In the lab-scale study, three fish (25%) demonstrated hepatic nuclear density in the control group, while no obvious abnormalities were observed in the fish fed the MsYbP (Figure 1B). Similarly, in the pilot-scale study, five fish (42%) exhibited nuclear density in the control group (Figure 1B), while only two fish (17%) showed hepatic nuclear density in the YANG group (Figure 1B).

### 3.3. Plasma Liver Functions

In both studies, the YANG group, consisting of fish fed MsYbP, exhibited a significant reduction in the plasma ALT and AST levels compared to those fed the control diet (Figure 2, *p* < 0.05). Moreover, the plasma AKP level showed a significant decrease in the fish fed MsYbP in the pilot-scale study, with no statistical differences observed in the lab-scale study (Figure 2, *p* > 0.05).

### 3.4. Plasma and Liver Antioxidation Parameters

In the lab-scale study, the fish fed the MsYbP showed lower plasma MDA levels compared to those fed the control diet (Figure 3A, *p* < 0.05), while no statistical difference was found in the pilot-scale study (Figure 3A, *p* > 0.05). Furthermore, the MsYbP-fed fish had significantly higher plasma T-AOC and T-SOD levels than the control-fed fish in both studies (Figure 3B,C, *p* < 0.05). There were no significant differences in the plasma GSH-Px and ROS levels between the dietary treatments in both studies (Figure 3D,E, *p* > 0.05).

Additionally, the fish in the YANG group had significantly decreased liver MDA levels in the lab-scale study (Figure 4A, *p* < 0.05). The MsYbP-fed fish had significantly increased liver T-AOC and GSH-Px levels compared to the control in both studies (Figure 4B,D, *p* < 0.05). There were no statistical differences in the liver T-SOD between the dietary treatments in either scale study (Figure 4C, *p* < 0.05). Higher hepatic CAT levels in the MsYbP-fed fish were only observed in the pilot-scale study (Figure 4E, *p* < 0.05).

### 3.5. Hepatic Antioxidation Relevant Genes Expression

In Figure 5, the MsYbP-fed fish showed significantly higher expression levels of the antioxidant-related gene *nrf2* in the pilot-scale study, but no statistical difference in the lab-scale study. Furthermore, in both studies, the fish in the YANG had markedly elevated gene expression levels of keap1 and other anti-oxidative enzymes-related genes (*cu/zn-sod*, *mn-sod*, *cat* and *gsh-px*) compared to the control (Figure 5, *p* < 0.05).

### 3.6. Hepatic Immune Relevant Gene Expression

In the lab-scale study, the fish fed MsYbP exhibited a significant increase in the expressions of inflammation-related genes, namely *tnfα*, *il-1β*, and *tgf-β1* (Figure 6, *p* < 0.05).

Conversely, in the pilot-scale study, the gene expressions of *tnfα* and *il-10* were suppressed in the fish in the YANG group (Figure 6, *p* < 0.05). Additionally, the fish in the YANG group demonstrated significantly lower hepatic *tnfα* and *il-10* contents compared to the control (Figure 6, *p* < 0.05).

## 4. Discussion

Our observations of higher growth performance in the YANG group in both the lab-scale and pilot-scale studies suggest an enhancement in growth when the fish were fed the MsYbP. Our results align with previous findings indicating that dietary yeast-based paraprobiotics can improve growth in various fish species, including rainbow trout [17], tilapia [14], hybrid sturgeon [18], common carp [19], and grass carp [20]. In our lab-scale study, a higher feed consumption was observed in the fish fed yeast-based paraprobiotics compared to those fed the control diet, indicating a better palatability of the YANG group and higher growth performance, as feed consumption is significantly influenced by palatability. This is in line with a previous finding in European seabass showing that the dietary inclusion of yeast-based paraprobiotics improved the growth performance by promoting the feed conversion ratio [21]. Furthermore, studies on European seabass and rainbow trout [21,23] have reported positive effects of MsYbP supplementation on intestinal morphometry, leading to increased nutrient absorption and subsequent improvements in growth performance. Our recent findings of enhanced intestinal permeability, peptide transport, and intestinal health of largemouth bass fed a cottonseed protein concentrate-based diet supplemented with MsYbP [22] may provide support for the higher growth observed in the current study. Another well-established explanation is that the yeast extract cytosolic fraction, containing bioactive components such as nucleotides and small peptides, has been observed to enhance the growth performance and gut health of fish through a more efficient metabolic use of nutrients [31]. It is worth noting that, due to the pilot-scale experiment conducted in a large-scale aquaculture farm, it was not feasible to accurately record the overall survival and feed consumption throughout the entire experimental phase. This limitation represents a regrettable aspect of the pilot-scale experiment. However, the experiment still holds significant value. Firstly, the sample size was sufficient to represent the overall effects. Secondly, most studies in aquaculture regarding functional additives lack long-term and pilot-scale experiments. Our experiment holds considerable practical significance for actual production. Overall, from the lab-scale to the pilot-scale studies, the dietary MsYbP consistently promoted fish growth, suggesting its potential as a profitable feed additive.

The observed growth performance is intricately linked to the liver health condition. In both the lab-scale and pilot-scale studies, the control-fed fish exhibited a higher ratio of the hepatic nuclear dense phenotype compared to the MsYbP-fed fish. This is consistent with previous research indicating that largemouth bass, as traditional carnivorous fish, may display a higher hepatocyte vacuolization index and lower growth when fed a formulated diet, as opposed to forage-fed fish [26]. In our study, the higher expression signal of cleaved caspase 3 was also observed in fish with the hepatic nuclear dense phenotype, signifying an active apoptotic response. This activated caspase response could induce cell death and stimulate cellular proliferation to repair the damaged liver [25]. Concurrently, our study revealed the significant suppression of liver functional indices, including the plasma AST, ALT, and AKP levels, in the fish in the YANG group compared to the control group. This aligns with several previous studies on largemouth bass [25,29,32] and Nile tilapia [33], which found that improved hepatocyte morphology correlated with a decrease in the plasma AST and ALT levels, attributed to the reduction of excess oxidative responses, which supports our findings.

Concerning the antioxidative capacity, it is well-documented that metabolic liver disease in largemouth bass is associated with abnormal lipid generation [25,26], leading to an increase in lipid peroxidative production, such as MDA [34,35]. In our lab-scale study, the fish fed MsYbP exhibited lower plasma MDA contents compared to the MsYbP-fed fish, supporting the notion of better liver health in the YANG group. However, no significant diet effects on the plasma and hepatic MDA levels were observed in the pilot-scale study, possibly influenced by factors such as the experimental conditions, fish age, and the duration of the experiment. Metabolic liver disease can induce an increase in several antioxidative enzymes, including T-SOD, GSH-Px, and CAT, enhancing antioxidative defense to eliminate excessive MDA content and protect the liver from peroxidative damage. Modulating oxidative stress through the regulation of antioxidant enzymes presents a potential therapeutic avenue for hepatic steatosis [36]. And the up-regulation of the antioxidant capacity could benefit the hepatic histomorphology, thus improving growth [37]. One previous study on largemouth seabass also reported that yeast supplementation could increase the hepatic T-SOD, GSH-Px, and CAT activities, positively affecting the hepatic histomorphology [32], which is in line with our findings showing that antioxidant relevant enzymes and genes were enhanced in the fish fed MsYbP. Furthermore, in our study, the fish fed the MsYbP supplementation showed significantly higher gene expressions of *nrf2* and *keap1*, which could participate in the activation of antioxidative genes [38]. The nrf2-keap1 pathway, as a key antioxidant response element recognizer, holds potential as a crucial target for preventing liver disease development [39]. Nrf2 activation induces the transcription of numerous antioxidative enzymes, thereby increasing the antioxidative capacity [40], aligning with our results. Its activation could also indirectly induce the up-regulation of *keap1* via the negative feedback regulation [41]. As the negative regulators, *keap1* could inactivate the *nrf2* binding the antioxidant response element and lead to E3 ubiquitination degradation [42]. Thus, the present study suggests that incorporating MsYbP into the diet increases the antioxidative capacity, benefiting growth and liver health.

Inflammatory responses have been proposed as potential indicators of liver health in largemouth bass [26]. In the present study, the dietary MsYbP tended to elevate the hepatic inflammation responses, including *tnfα*, *il-1β*, and *tgf-β1*, in the lab-scale study but suppressed these responses in the pilot-scale study. Similarly, one previous study on largemouth bass found that fish fed a yeast-based prebiotic diet could stimulate the up-regulation of genes related to hepatic inflammation responses in a short-term duration (2 weeks), but down-regulated their responses in a long-term duration (8 weeks) [29]. Yu and colleagues suggested that the short-term up-regulation of mRNA levels in inflammation responses might be associated with excessive fat accumulation and oxidative stress [29]. However, in our lab-scale study, dietary MsYbP mitigated liver oxidative stress and improved liver health, as discussed earlier. The up-regulation of inflammation-relevant genes in our study may be explained by the yeast extract playing a crucial role as an immuno-stimulator, activating the hepatic expression of *tnfα*, *il-1β*, and *tgf-β1* under laboratory conditions [13,21,23]. The immune-stimulating effects of dietary MsYbP are likely due to the presence of various bioactive substances such as antimicrobial peptides, β-glucans, mannan oligosaccharides, and nucleotides, which can increase the expression levels of inflammatory cytokines [13]. Conversely, in the pilot-scale study, lower inflammation responses were observed in the YANG group, alongside higher growth performance, antioxidative capacity, and improved liver health, showing the beneficial effects of dietary MsYbP. Inflammation responses reflect a physiological homeostatic process, involving the mutual balance of pro- and anti-inflammatory factors [43]. The discrepancies in the results between the lab-scale and pilot-scale studies may be attributed to the different host physiological conditions arising from variations in the experimental conditions, the supplementation levels of MsYbP, fish age, and duration. Future research should delve into whether the observed beneficial effects on growth and liver health induced by dietary MsYbP in the pilot-scale investigation are linked to the down-regulated inflammatory responses.

## 5. Conclusions

The incorporation of multi-strain yeast-based paraprobiotics in the diet of largemouth bass demonstrated a significant potential to enhance their growth. Furthermore, dietary MsYbP augmented the hepatic antioxidation capacity by reducing the malondialdehyde levels and elevating the levels or gene expressions related to specific antioxidative markers. These antioxidation responses elicited by dietary MsYbP influenced the hepatic histomorphology and improved the liver health of largemouth bass. In conclusion, the implications of our study extend to the aquaculture industry, providing valuable insights into the use of yeast-based paraprobiotics as a nutritional supplement to enhance the antioxidative capacity and liver health of farmed fish.

## Figures and Tables

**Figure 1 antioxidants-13-00792-f001:**
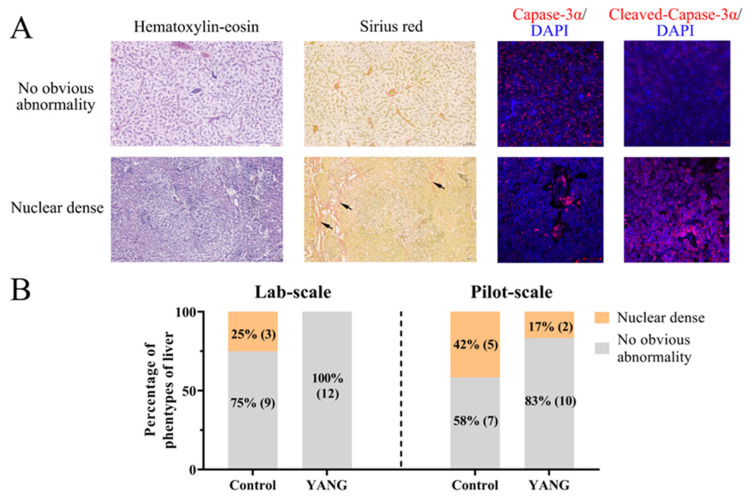
The hepatic histopathological examination of largemouth bass in the lab-scale and pilot-scale studies. (**A**) The hematoxylin–eosin, Sirius Red (the red collagen fibers of liver marked by black arrow) and immunofluorescence (caspase 3 and cleaved-caspase 3; DAPI: 4′,6-diamidino-2-phenylindole) staining of two hepatic phenotypes. (**B**) The hepatic phenotypes in the lab-scale and pilot-scale studies (n = 12).

**Figure 2 antioxidants-13-00792-f002:**
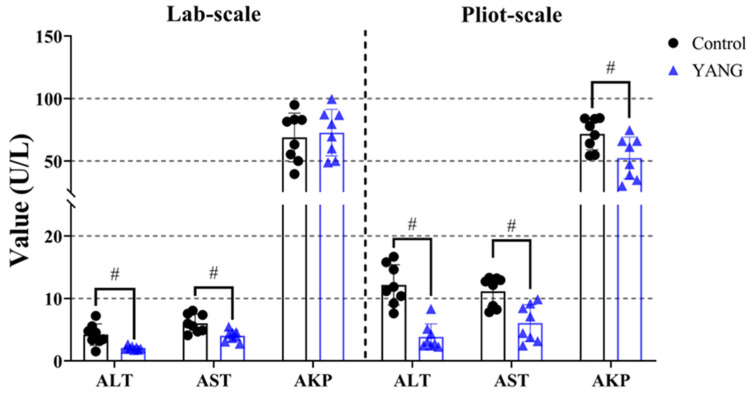
Effects of dietary MsYbP on plasma biochemical parameters of largemouth bass in the lab-scale and pilot-scale studies (n = 8, independent *t*-test, means ± SEM). ALT: alanine aminotransferase; AST: aspartate aminotransferase; AKP: alkaline phosphatase. Values with hashtag (^#^) are significantly lower compared to the control group (*p* < 0.05).

**Figure 3 antioxidants-13-00792-f003:**
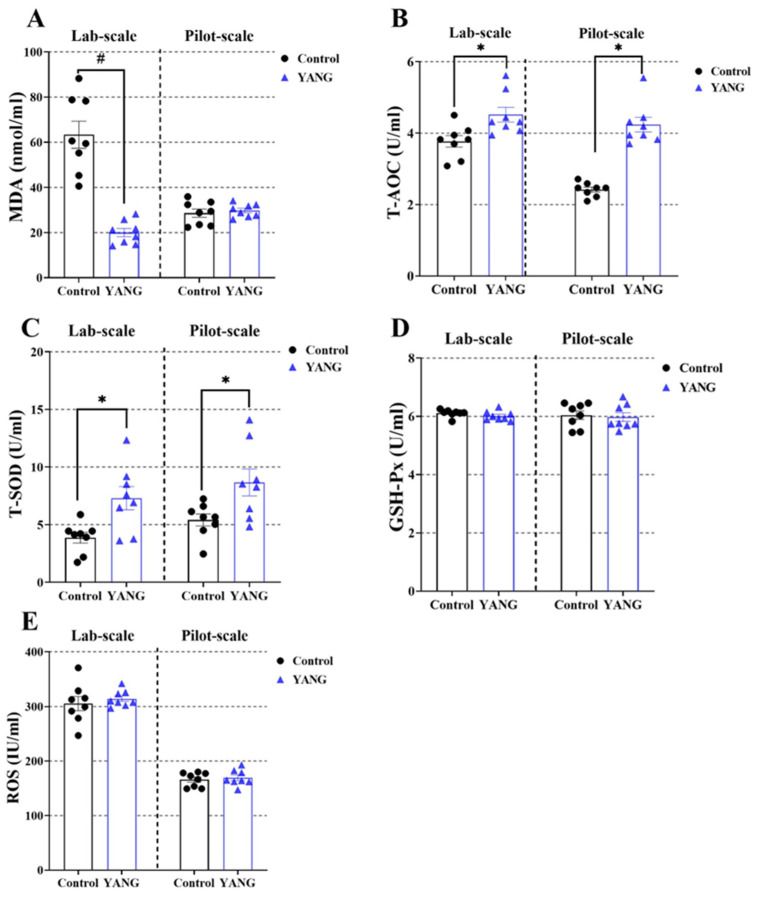
Effects of dietary MsYbP on plasma antioxidative parameters of largemouth bass in the lab-scale and pilot-scale studies (n = 8; independent *t*-test; means ± SEM). (**A**): malondialdehyde (MDA); (**B**): total antioxidant capacity (T-AOC); (**C**): total superoxide dis-mutase (T-SOD); (**D**): glutathione peroxidase (GSH-Px); (**E**): reactive oxygen species (ROS); Values with hashtag (^#^) and asterisk (*) are significantly lower or higher compared to the control, respectively (*p* < 0.05).

**Figure 4 antioxidants-13-00792-f004:**
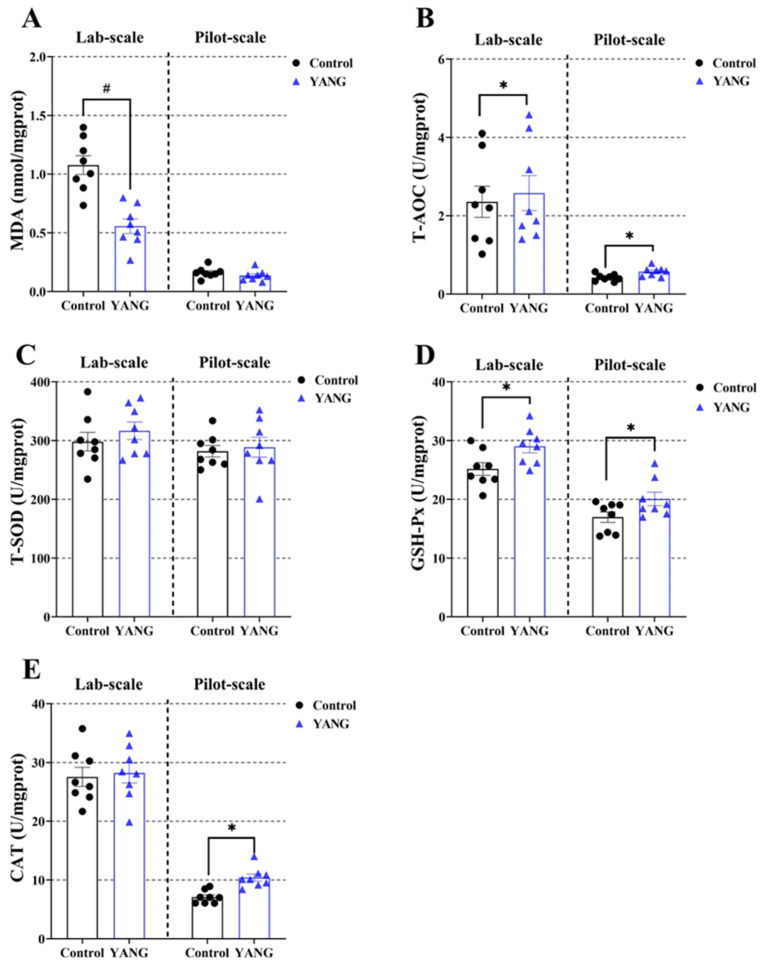
Effects of dietary MsYbP on hepatic antioxidative parameters of largemouth bass in the lab-scale and pilot-scale studies (n = 8; independent *t*-test; means ± SEM). (**A**): malondialdehyde (MDA); (**B**): total antioxidant capacity (T-AOC); (**C**): total superoxide dis-mutase (T-SOD); (**D**): glutathione peroxidase (GSH-Px); (**E**): catalase (CAT); Values with hashtag (^#^) and asterisk (*) are significantly lower or higher compared to the control, respectively (*p* < 0.05).

**Figure 5 antioxidants-13-00792-f005:**
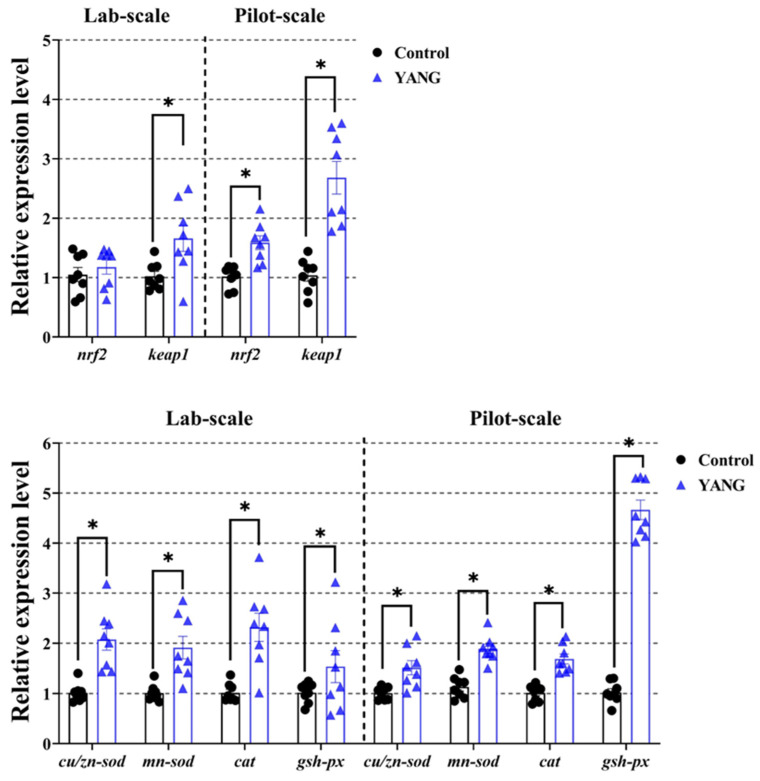
Effects of dietary MsYbP on hepatic antioxidative mRNA expression of largemouth bass in the lab-scale and pilot-scale studies (n = 8; independent *t*-test; means ± SEM). Values with asterisk (*) are significantly higher compared to the control (*p* < 0.05).

**Figure 6 antioxidants-13-00792-f006:**
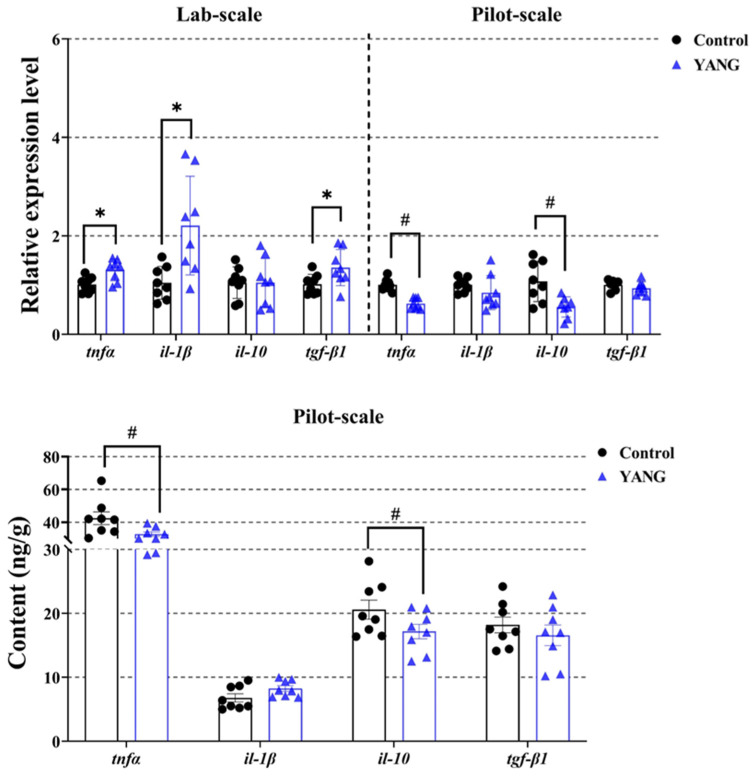
Effects of dietary MsYbP on hepatic inflammatory mRNA expression and factor contents of largemouth bass in the lab-scale and pilot-scale studies (n = 8; independent *t*-test; means ± SEM). Values with hashtag (^#^) and asterisk (*) are significantly lower or higher compared to the control, respectively (*p* < 0.05).

**Table 1 antioxidants-13-00792-t001:** Compositions of diets in the lab-scale experiment (g/100 g).

Diets	Control	YANG
Ingredients		
Fishmeal ^1^	30	30
Cottonseed protein concentrate ^2^	23.5	23.5
*Clostridium autoethanogenum* protein ^3^	4	4
Wheat flour	9	9
Tapioca starch	5	5
α-cellulose	4.6	4.6
Spray-dried blood cell powder	4	4
Wheat gluten	4	4
Soybean meal ^4^	2	2
Monocalcium phosphate Ca(H_2_PO_4_)_2_	1.7	1.7
Kelp powder	1.5	1.5
Fish oil	3.52	3.52
Plant oil mix	5.5	5.5
Vitamin and mineral premix ^5^	1.38	1.38
DL-Methionine	0.2	0.2
L-Threonine	0.1	0.1
MsYbP ^6^	0	0.08
Proximate composition (DM, %)		
Dry matter	93.9	92.6
Crude protein	50.8	51.1
Crude lipid	12.1	12.3
Crude ash	10.1	10.0
Gross energy (MJ/kg)	22.1	21.9

^1^ The fishmeal was purchased from Tecnol’ogica de Alimentos S.A., Ltd. (Lima, Peru); 67.0% crude protein and 10.1% crude lipid. ^2^ The cottonseed protein concentrate was purchased from Xinjiang Jinlan Vegetable Protein Co., Ltd. (Uygur Zizhiqu, China); 65.3% crude protein and 0.5% crude lipid. ^3^ The Clostridium autoethanogenum protein was purchased from Beijing Shoulang Biotechnology Co., Ltd., Beijing (Beijing, China); 81.2% crude protein and 0.3% crude lipid. ^4^ The soybean meal was purchased Yihai Kerry Investment Co., Ltd. (Shanghai, China); 46.6% crude protein and 0.55% crude lipid. ^5^ The detailed information of the vitamin and mineral premix (mg/kg diet) can be found in our previously published study [23].^6^ The MsYbP (multi-strain yeast-based paraprobiotic), consisting of inactive cells and polysaccharides (β-glucan, mannan oligosaccharides, and other oligosaccharides) from *Saccharomyces cerevisiae* and *Cyberlindnera jadinii*, was supplied by Lallemand SAS (Blagnac, France).

**Table 2 antioxidants-13-00792-t002:** Primer sequences for real-time PCR.

Gene	Primer	Sequence 5′–3′	Amplicon Size (bp)	Amplification Efficiency (%)	Tm (°C)
*ef-1α*	F	TGCTGCTGGTGTTGGTGAGTT	147	102.8	60.4
	R	TTCTGGCTGTAAGGGGGCTC			
*nrf2*	F	CAGACGGGGAAACAAACAATG	170	96.8	55.0
	R	GGGGTAAAATACGCCACAATAAC			
*keap1*	F	TCATTGGGGAATCACATCTTTG	199	99.6	56.5
	R	TGTCCAGAAAAGTGTTGCCATC			
*cu/zn-sod*	F	ACCACAGAAACTTACGCGACA			
	R	TTCACAGGGTCTGAATCGCC			
*mn-sod*	F	TGCTTTGCAGAGTTGGTCAG	183	97.0	58.0
	R	CATAAGTGGCATGGTGCTTG			
*cat*	F	GTCCTTCATCCACTCCCAGA	162	90.0	55.0
	R	CCATAGCCATTCATGTGACG			
*gsh-px*	F	CTCCTCAACCAGGCAAAC			
	R	ATACCCCCCTCACAACAA			
*tnfα*	F	CTTCGTCTACAGCCAGGCATCG	161	105.7	63.0
	R	TTTGGCACACCGACCTCACC			
*il-1β*	F	CGTGACTGACAGCAAAAAGAGG	166	101.3	59.4
	R	GATGCCCAGAGCCACAGTTC			
*il-10*	F	CGGCACAGAAATCCCAGAGC	119	113.6	62.1
	R	CAGCAGGCTCACAAAATAAACATCT			
*tgf-β1*	F	GCTCAAAGAGAGCGAGGATG	118	95.6	59.0
	R	TCCTCTACCATTCGCAATCC			

*ef-1α*, elongation factor 1α; *nrf2*, nuclear factor erythroid 2-related factor 2; *keap1*, kelch-1ike ech-associated protein; *cu/zn-sod*, cu/zn superoxide dismutase; *mn-sod*, mn superoxide dismutase; *cat*, catalase; *gsh-px*, glutathione peroxidase; *tnfα*, tumor necrosis factor α; *il-1β*, interleukin-1β; *il-10*, interleukin-10; *tgf-β1*, transforming growth factor β1; F, forward primer; R, reverse primer.

**Table 3 antioxidants-13-00792-t003:** Effects of MsYbP on growth, morphometry, and economic indices of largemouth bass.

	Lab-Scale			Pilot-Scale		
	Control	YANG	*p* Value	Control	YANG	*p* Value
Growth (lab-scale: n = 4 tank; pilot-scale: n = 12 fish; mean ± SEM)						
FBW (g) ^a^	102.4 ± 1.6	116.8 ± 1.3	0.001	323.7 ± 10.8	367.7 ± 7.0	0.04
WGR (%) ^b^	213.9 ± 3.3	272.7 ± 5.0	0.001	243.7 ± 10.8	287.7 ± 7.0	0.04
Survival (%) ^c^	92.5 ± 3.3	100.0 ± 0.0	0.06	-	-	-
Feed consumption (g) ^c^	1443 ± 11	1698 ± 24	0.001	-	-	-
Morphometry (lab-scale: n = 13–16 fish; pilot-scale: n = 12 fish; mean ± SEM)						
CF ^d^	2.57 ± 0.06	2.60 ± 0.06	0.66	1.73 ± 0.05	1.81 ± 0.04	0.19
VSI (%) ^e^	6.82 ± 0.18	6.92 ± 0.28	0.73	10.47 ± 0.38	10.17 ± 0.38	0.59
HSI (%) ^f^	1.55 ± 0.07	1.50 ± 0.08	0.61	2.47 ± 0.098	2.58 ± 0.09	0.43
VAI (%) ^g^	1.59 ± 0.18	2.07 ± 0.19	0.04	2.09 ± 0.12	2.63 ± 0.21	0.03

^a^ FBW: final average body weight; ^b^ WGR (weight gain rate; %) = 100 × ((final body weight − initial body weight)/initial body weight); ^c^ survival rates and feed consumption in lab-scale study are presented as described in our previous study [22], while the data on survival rates and feed consumption are not available in the pilot-scale study because they are the intellectual property of a commercial enterprise; ^d^ CF (condition factor) = 100 × (body weight; g)/(body length^3^; cm); ^e^ VSI (viscero-somatic index; %) = 100 × (viscera weight/body weight); ^f^ HSI (hepatosomatic index; %) = 100 × (liver weight)/body weight; ^g^ VAI (visceral adipose index; %) = 100 × visceral adipose weight/body weight. -: no data.

## Data Availability

The authors confirm that the data supporting the findings of this study are included within the article.

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
