# Peer review of "Growth, Antioxidant Capacity, and Liver Health in Largemouth Bass (Micropterus salmoides) Fed Multi-Strain Yeast-Based Paraprobiotic: A Lab-to-Pilot Scale Evaluation"

_antioxidants, 2024, doi:10.3390/antiox13070792_

Round 1
Reviewer 1 Report
Comments
“Growth, Antioxidant capacity and Liver health in Largemouth 2 Bass (Micropterus salmoides) Fed Multi-Strain Yeast-Based 3 Paraprobiotic: A Lab-to-Pilot Scale Evaluation”
This study assessed the impact of MsYbP in largemouth bass growth, anti-15 oxidant capacities, immune responses, and liver health through lab-scale (65 days) and pilot-scale 16 (15 weeks) experiments. The concentrations used were 0.08% and 0.1% MsYbP in lab-scale and pilot-scale. In the lab-scale study 20 fish per replicate were used (n=4). While in a pilot scale (n=3) with 1500 fish of 80 gr was used. The study conclusion emphasizes the significance of employing yeast-based 33 paraprobiotics in commercial conditions. However, my principal concern with this study is why the authors use different % of supplementation in lab-scale and pilot-scale? My question is, maybe the differences in the gen expression, oxidative stress and so on were for the difference in the concentration percent?
What means? ALT? AST? Etc Fig 2 titles….
w
Author Response
Review 1
Major comments
“Growth, Antioxidant capacity and Liver health in Largemouth Bass (Micropterus salmoides) Fed Multi-Strain Yeast-Based Paraprobiotic: A Lab-to-Pilot Scale Evaluation” This study assessed the impact of MsYbP in largemouth bass growth, anti-oxidant capacities, immune responses, and liver health through lab-scale (65 days) and pilot-scale 16 (15 weeks) experiments. The concentrations used were 0.08% and 0.1% MsYbP in lab-scale and pilot-scale. In the lab-scale study 20 fish per replicate were used (n=4). While in a pilot scale (n=3) with 1500 fish of 80 gr was used. The study conclusion emphasizes the significance of employing yeast-based paraprobiotics in commercial conditions.
- However, my principal concern with this study is why the authors use different % of supplementation in lab-scale and pilot-scale? My question is, maybe the differences in the gene expression, oxidative stress and so on were for the difference in the concentration percent?
Responses: In lab-scale study, the initial body weight was approximately 31g, and the level of MsYbP was determined according to our previous experimental references (Xie et al., animal nutrition, 2023; Xie et al., aquaculture, 2022). In the pilot-scale experiments, the initial body weight was 80g, and the cultivation period was 15 weeks, so the level was based on a weight increase to 1000 mg/kg. Although the supplemented level differed between the two studies, each study had a corresponding control group. We observed that growth, antioxidant capacity, and liver health showed the same trends in both scale experiments. However, there were different trends in the expression of certain inflammation-related markers, which may be a contributing factor. I will add a discussion on this aspect, as line 383-385: “The discrepancy of results between the lab-scale and pilot-scale studies may be attributed to different host physiological conditions arising from variations in experimental conditions, the supplementation level of MsYbP, fish age, and duration.”
References:
Xie, X., Liang, X., Wang, H., Zhu, Q., Wang, J., Chang, Y., Leclercq, E., Xue, M., Wang, J., Effects of paraprobiotics on bile acid metabolism and liver health in largemouth bass (Micropterus salmoides) fed a cottonseed protein concentrate-based diet, Anim Nutr 2023, 13, 302-312.
Xie, X., Wang, J., Guan, Y., Xing, S., Liang, X., Xue, M., Wang, J., Chang, Y., Leclercq, E., Cottonseed protein concentrate as fishmeal alternative for largemouth bass (Micropterus salmoides) supplemented a yeast-based paraprobiotic: Effects on growth performance, gut health and microbiome, Aquaculture 2022, 551, 737898.
- What means? ALT? AST? Etc Fig 2 titles….
Responses: The related results have already been introduced in Lines 160-161, and we will also explain these results in detail under the legend of Fig. 2.

Reviewer 2 Report
Growth, Antioxidant capacity and Liver health in Largemouth Bass (Micropterus salmoides) Fed Multi-Strain Yeast-Based Paraprobiotic: A Lab-to-Pilot Scale Evaluation. The study reported that dietary multi-strain yeast-based paraprobiotic in enhancing the growth and liver health of largemouth bass, potentially through increased antioxidative capacity and modulation of immune responses in lab-scale and pilot-scale experiments. Paraprobiotics plays a significant role in commercial conditions.
The study is well-structured, and the experimental design is robust, addressing a critical issue. However, there are several areas where the manuscript could be improved for clarity, depth, and overall scientific rigor.
The following comments need to be reviewed and addressed by the authors:
1. Revise the manuscript to correct grammatical errors and improve sentence structure for better clarity. A thorough proofreading could be beneficial.
2. Check the Journal format specially at the reference section.
3. Add P value to the abstract, revise the abstract.
4. Augment the introduction with more recent citations to strengthen the background information.
5. Clearly delineate the research gaps this study seeks to fill within the introduction.
6. Clearly state the hypothesis being tested within the introduction section.
7. Strengthen the conclusion by more explicitly linking the results to the broader implications. Suggest practical applications of the findings and potential for future research.
L22 & 24: add p value.
L57-60: repetition, reorganize the paragraph
L65-67: repetition, reorganize the paragraph
Table 1: analyzed values for all diets are required specially for key nutrients such as crude protein
L190-191: rewire.
Table 3: add P value for example 0.001 not 0.00.
Fif. 2: there is no * used.
Author Response
Major comments
Growth, Antioxidant capacity and Liver health in Largemouth Bass (Micropterus salmoides) Fed Multi-Strain Yeast-Based Paraprobiotic: A Lab-to-Pilot Scale Evaluation. The study reported that dietary multi-strain yeast-based paraprobiotic in enhancing the growth and liver health of largemouth bass, potentially through increased antioxidative capacity and modulation of immune responses in lab-scale and pilot-scale experiments. Paraprobiotics plays a significant role in commercial conditions.
The study is well-structured, and the experimental design is robust, addressing a critical issue. However, there are several areas where the manuscript could be improved for clarity, depth, and overall scientific rigor.
The following comments need to be reviewed and addressed by the authors:
- Revise the manuscript to correct grammatical errors and improve sentence structure for better clarity. A thorough proofreading could be beneficial.
Responses: We have had a native English-speaking colleague review and correct the whole content.
- Check the Journal format specially at the reference section.
Responses: I have revised the reference as requested. Please see the latest version.
“FAO, FAO Yearbook Fishery and Aquaculture Statistics 2022; Food and agriculture organization of the United Nations: Rome, Italy,p. 10.
Loc, V.T.T.; Bush, S.R.; Sinh, L.X.; Khiem, N.T. High and low value fish chains in the Mekong Delta: challenges for livelihoods and governance. Environment, development and sustainability 2010, 12, 889-908.
Hoseinifar, S.H.; Ringø, E.; Shenavar Masouleh, A.; Esteban, M.Á. Probiotic, prebiotic and synbiotic supplements in sturgeon aquaculture: a review. Reviews in Aquaculture 2016, 8, 89-102.
Liu, A.; Santigosa, E.; Dumas, A.; Hernandez, J.M. Vitamin nutrition in salmonid aquaculture: From avoiding deficiencies to enhancing functionalities. Aquaculture 2022, 561, 738-654.
Dawood, M.A.; Koshio, S.; Esteban, M.Á. Beneficial roles of feed additives as immunostimulants in aquaculture: a review. Reviews in Aquaculture 2018, 10, 950-974.
Taverniti, V.; Guglielmetti, S. The immunomodulatory properties of probiotic microorganisms beyond their viability (ghost probiotics: proposal of paraprobiotic concept). Genes & Nutrition 2011, 6, 261-274.
Li, S.; Tran, N.T. Paraprobiotics in Aquaculture, Probiotics in Aquaculture, Springer2022, pp. 131-164.
Gyan, W.R.; Ayiku, S.; Yang, J.; Asumah, J. Effects of yeast antimicrobial peptide in aquaculture. Journal of Fisheries and Aquaculture Development 2019, 6, 2577-1493.
Petit, J.; Wiegertjes, G.F.; Long-lived effects of administering beta-glucans: Indications for trained immunity in fish. Developmental & Comparative Immunology 2016, 64, 93-102.
Petit, J.; Bailey, E.C.; Wheeler, R.T.; De Oliveira, C.A.; Forlenza, M.; Wiegertjes, G.F. Studies into β-glucan recognition in fish suggests a key role for the C-type lectin pathway. Frontiers in immunology 2019, 10, 280.
Korcova, J.; Machova, E.; Filip, J.; Bystricky, S. Biophysical properties of carboxymethyl derivatives of mannan and dextran. Carbohydrate polymers 2015, 134, 6-11.
Cummings, R.D. The mannose receptor ligands and the macrophage glycome. Current opinion in structural biology 2022, 75, 102394.
Wang, J.; Lei, P.; Gamil, A.A.A.; Lagos, L.; Yue, Y.; Schirmer, K.; Mydland, L.T.; Overland, M.; Krogdahl, Å.; Kortner, T.M. Rainbow Trout (Oncorhynchus Mykiss) Intestinal Epithelial Cells as a Model for Studying Gut Immune Function and Effects of Functional Feed Ingredients. Front Immunol 2019, 10, 152.
Divya, M.; Gopi, N.; Iswarya, A.; Govindarajan, M.; Alharbi, N.S.; Kadaikunnan, S.; Khaled, J.M.; Almanaa, T.N.; Vaseeharan, B. beta-glucan extracted from eukaryotic single-celled microorganism Saccharomyces cerevisiae: Dietary supplementation and enhanced ammonia stress tolerance on Oreochromis mossambicus. Microb Pathog 2020, 139, 103917.
Dimitroglou, A.; Merrifield, D.L.; Moate, R.; Davies, S.J.; Spring, P.; Sweetman, J.; Bradley, G. Dietary mannan oligosaccharide supplementation modulates intestinal microbial ecology and improves gut morphology of rainbow trout, Oncorhynchus mykiss (Walbaum). Journal of animal science 2009, 87, 3226-34.
Faustino, M.; Durao, J.; Pereira, C.F.; Pintado, M.E.; Carvalho, A.P. Mannans and mannan oligosaccharides (MOS) from Saccharomyces cerevisiae-A sustainable source of functional ingredients. Carbohydrate Polymers 2021, 272, 118467.
Villumsen, K.R.; Ohtani, M.; Forberg, T.; Aasum, E.; Tinsley, J.; Bojesen, A.M. Synbiotic feed supplementation significantly improves lipid utilization and shows discrete effects on disease resistance in rainbow trout (Oncorhynchus mykiss). Sci Rep 2020, 10, 16993.
Wu, X.; Hao, Q.; Teame, T.; Ding, Q.; Liu, H.; Ran, C.; Yang, Y.; Xia, L.; Wei, S.; Zhou, Z.; Zhang, Z.; Zhang, Y. Gut microbiota induced by dietary GWF® contributes to growth promotion, immune regulation and disease resistance in hybrid sturgeon (Acipenserbaerii x Acipenserschrenckii): Insights from a germ-free zebrafish model. Aquaculture 2020, 520, 734966.
Meng, D.; Hao, Q.; Zhang, Q.; Yu, Z.; Liu, S.; Yang, Y.; Ran, C.; Zhang, Z.; Zhou, Z. A compound of paraprobiotic and postbiotic derived from autochthonous microorganisms improved growth performance, epidermal mucus, liver and gut health and gut microbiota of common carp (Cyprinus carpio). Aquaculture 2023, 570.
Huang, W.; Xiao, X.; Hu, W.; Tang, T.; Bai, J.; Zhao, S.; Ao, Z.; Wei, Z.; Gao, W.; Zhang, W. Effects of dietary nucleotide and yeast cell wall on growth performance, feed utilization, anti-oxidative and immune response of grass carp (Ctenopharyngodon idella). Fish Shellfish Immunol 2023, 108574.
Rawling, M.D.; Pontefract, N.; Rodiles, A.; Anagnostara, I.; Leclercq, E.; Schiavone, M.; Castex, M.; Merrifield, D.L. The effect of feeding a novel multistrain yeast fraction on European seabass (Dicentrachus labrax) intestinal health and growth performance. Journal of the World Aquaculture Society 2019, 50, 1108-1122.
Xie, X.; Wang, J.; Guan, Y.; Xing, S.; Liang, X.; Xue, M.; Wang, J.; Chang, Y.; Leclercq, E. Cottonseed protein concentrate as fishmeal alternative for largemouth bass (Micropterus salmoides) supplemented a yeast-based paraprobiotic: Effects on growth performance, gut health and microbiome. Aquaculture 2022, 551, 737898.
Rawling, M.; Leclercq, E.; Foey, A.; Castex, M.; Merrifield, D. A novel dietary multi-strain yeast fraction modulates intestinal toll-like-receptor signalling and mucosal responses of rainbow trout (Oncorhynchus mykiss). PLoS One 2021, 16, e0245021.
Xie, X.; Liang, X.; Wang, H.; Zhu, Q.; Wang, J.; Chang, Y.; Leclercq, E.; Xue, M.; Wang, J. Effects of paraprobiotics on bile acid metabolism and liver health in largemouth bass (Micropterus salmoides) fed a cottonseed protein concentrate-based diet. Anim Nutr 2023, 13, 302-312.
Yu, H.H. Metabolic Liver Disease Induced by a High Carbohydrate Diet in Largemouth Bass (Micropterus salmoides), and the Alleviating Mechanism of Bile Acid Supplementation, Chinese Academy of Agricultural Sciences, Ph.D thesis, 2019.
Ma, D.M.; Fan, J.J.; Zhu, H.P.; Su, H.H.; Jiang, P.; Yu, L.Y.; Liao, G.L.; Bai, J.J. Histologic examination and transcriptome analysis uncovered liver damage in largemouth bass from formulated diets. Aquaculture 2020, 526, 735329.
Amoah, A.; Coyle, S.D.; Webster, C.D.; Durborow, R.M.; Bright, L.A.; Tidwell, J.H. Effects of graded levels of carbohydrate on growth and survival of largemouth bass, Micropterus salmoides. Journal of the World Aquaculture Society 2008, 39, 397-405.
Xu, X.; Chen, N.; Liu, Z.; Gou, S.; Yin, J. Effects of dietary starch sources and levels on liver histology in largemouth bass, Micropterus salmoides. Journal of Shanghai Ocean University 2016, 25, 61-70.
Yu, H.H.; Liang, X.F.; Chen, P.; Wu, X.F.; Zheng, Y.H.; Luo, L.; Qin, Y.C.; Long, X.C.; Xue, M. Dietary supplementation of Grobiotic®-A increases short-term inflammatory responses and improves long-term growth performance and liver health in largemouth bass (Micropterus salmoides). Aquaculture 2019, 500, 327-337.
Pfaffl, M.W. A new mathematical model for relative quantification in real-time RT–PCR. Nucleic acids research 2001, 29, e45-e45.
Liu, D.; Ding, L.; Sun, J.; Boussetta, N.; Vorobiev, E. Yeast cell disruption strategies for recovery of intracellular bio-active compounds — A review. Innovative Food Science & Emerging Technologies 2016, 36, 181-192.
Xv, Z.; Zhong, Y.; Wei, Y.; Zhang, T.; Zhou, W.; Jiang, Y.; Chen, Y.; Lin, S. Yeast culture supplementation alters the performance and health status of juvenile largemouth bass (Micropterus salmoides) fed a high‐plant protein diet. Aquaculture Nutrition 2021, 27, 2637-2650.
Liu, Y.; Han, S.-L.; Luo, Y.; Li, L.-Y.; Chen, L.-Q.; Zhang, M.-L.; Du, Z.-Y. Impaired peroxisomal fat oxidation induces hepatic lipid accumulation and oxidative damage in Nile tilapia. Fish physiology and biochemistry 2020, 46, 1229-1242.
Morita, M.; Ishida, N.; Uchiyama, K.; Yamaguchi, K.; Itoh, Y.; Shichiri, M.; Yoshida, Y.; Hagihara, Y.; Naito, Y.; Yoshikawa, T.; Niki, E. Fatty liver induced by free radicals and lipid peroxidation. Free radical research 2012, 46, 758-65.
Ding, R.B.; Tian, K.; Cao, Y.W.; Bao, J.L.; Wang, M.; He, C.W.; Hu, Y.J.; Su, H.X.; Wan, J.B. Protective Effect of Panax notoginseng Saponins on Acute Ethanol-Induced Liver Injury Is Associated with Ameliorating Hepatic Lipid Accumulation and Reducing Ethanol-Mediated Oxidative Stress. Journal of Agricultural and Food Chemistry 2015, 63, 2413-2422.
Yang, J.-P.; Shin, J.-H.; Seo, S.-H.; Kim, S.-G.; Lee, S.H.; Shin, E.-H. Effects of antioxidants in reducing accumulation of fat in hepatocyte. International journal of molecular sciences 2018, 19, 2563.
Liu, J.; Xue, M.; Morais, S.; He, M.; Wang, H.; Wang, J.; Pastor, J.J.; Gonçalves, R.A.; Liang, X. Effects of a Phytogenic Supplement Containing Olive By-Product and Green Tea Extracts on Growth Performance, Lipid Metabolism, and Hepatic Antioxidant Capacity in Largemouth Bass (Micropterus salmoides) Fed a High Soybean Meal Diet. Antioxidants 2022, 11, 2415.
Cardozo, L.F.; Pedruzzi, L.M.; Stenvinkel, P.; Stockler-Pinto, M.B.; Daleprane, J.B.; Leite, M.; Jr.; Mafra, D. Nutritional strategies to modulate inflammation and oxidative stress pathways via activation of the master antioxidant switch Nrf2. Biochimie 2013, 95, 1525-33.
Xu, D.; Xu, M.; Jeong, S.; Qian, Y.; Wu, H.; Xia, Q.; Kong, X. The Role of Nrf2 in Liver Disease: Novel Molecular Mechanisms and Therapeutic Approaches. Frontiers in pharmacology 2018, 9, 1428.
Kobayashi, M.; Yamamoto, M. Molecular mechanisms activating the Nrf2-Keap1 pathway of antioxidant gene regulation. Antioxidants & redox signaling 2005, 7, 385-94.
Li, R.; Jia, Z.; Zhu, H. Regulation of Nrf2 signaling. Reactive oxygen species (Apex, NC) 2019,8, 312.
Zhang, D.D.; Lo, S.C.; Cross, J.V.; Templeton, D.J.; Hannink, M. Keap1 is a redox-regulated substrate adaptor protein for a Cul3-dependent ubiquitin ligase complex. Molecular and Cellular Biology 2004, 24, 10941-10953.
Kotas, Maya E.; Medzhitov, R. Homeostasis, Inflammation, and Disease Susceptibility. Cell 2015, 160, 816-827."
- Add P value to the abstract, revise the abstract.
Responses: we have added “P” and revised the abstract as suggested.
“The multi-strain yeast-based paraprobiotic (MsYbP) comprises inactive cells and polysaccharides (β-glucan, mannan-oligosaccharides and oligosaccharides) derived from Saccharomyces cerevisiae and Cyberlindnera jadinii, which could secure optimal growth and health in farmed fish. This study assessed the impact of MsYbP on the growth, immune responses, antioxidant capacities and liver health in largemouth bass (Micropterus salmoides) through lab-scale (65 days) and pilot-scale (15 weeks) experiments. Two groups of fish were monitored: one fed a control diet without MsYbP and another with 0.08% and 0.1% MsYbP in lab-scale and pilot-scale studies, respectively (referred to as YANG). In the lab-scale study, four replicates were conducted with 20 fish per replicate (average initial body weight = 31.0 ± 0.8g), while the pilot-scale study involved three replicates with approximately 1500 fish per replicate (average initial body weight = 80.0 ± 2.2g). The results indicated that MsYbP-fed fish exhibited a significant increase in growth performance in both studies (P < 0.05). Additionally, dietary MsYbP led to a noteworthy reduction in liver function parameters (ALT, AST, and AKP, P < 0.05) and hepatic nuclear density, indicating improved liver health. Furthermore, dietary MsYbP elevated antioxidative capacity by reducing malondialdehyde levels and increasing levels and gene expressions related to antioxidative markers (e.g., T-AOC, SOD, GSH-Px, CAT, nrf2, and keap1) in both studies (P < 0.05). In terms of hepatic immune responses, the lab-scale study showed an increase in inflammation-related gene expressions (e.g., il-1β and tgf-β1), while the pilot-scale study significantly suppressed expressions of genes related to inflammatory response (e.g., tnfα and il-10) (P < 0.05). In summary, our findings underscore the role of dietary multi-strain yeast-based paraprobiotic in enhancing the growth and liver health of largemouth bass, potentially through increased antioxidative capacity and modulation of immune responses, which emphasize the significance of employing yeast-based paraprobiotics in commercial conditions.”
- Augment the introduction with more recent citations to strengthen the background information.
Responses: I am a bit confused about this. This introduction cites a total of 29 articles, 20 of which were published in the last five years and include cutting-edge research on yeast-based paraprobiotics. Could you please specify which additional research background you would like me to include?
- Clearly delineate the research gaps this study seeks to fill within the introduction.
Responses: As suggested, we have delineated the research gaps in the introduction of this study, as below:
Line 76-78 :However, these relevant studies are primarily small-scale and short-term. There are no large-scale, long-term pilot studies, and there is limited research on antioxidants and liver health.
Line 82-87: Compared to forage fish, formulated diets for largemouth bass have posed challenges, such as the risk of inducing metabolic liver disease [26], as largemouth bass has a low tolerance for carbohydrates in the formulated diet[27, 28]. To address this issue, the supplementation of yeast-based paraprobiotic in the formulated diet was widely used to enhance the efficacy and alleviate potential adverse impact on the liver health of largemouth bass [29].
- Clearly state the hypothesis being tested within the introduction section.
Responses: As suggested, the hypothesis has been added in the introduction section, as below:
Line 88-91: Hence, our study aims to assess the effects of dietary multi-strain yeast-based paraprobiotics on the growth performance, immune responses, antioxidant capacities of largemouth bass in lab-scale and pilot-scale studies, and whether yeast-based paraprobioticsm could improve liver health of largemouth bass fed the formulated diet.
- Strengthen the conclusion by more explicitly linking the results to the broader implications. Suggest practical applications of the findings and potential for future research.
Responses: As suggested, we have revised the conclusion, as below:
Line 390-398: The incorporation of multi-strain yeast-based paraprobiotics in the diet of largemouth bass demonstrates a significant potential to enhance growth performance. Furthermore, dietary MsYbP augments hepatic antioxidation capacity by reducing malondialdehyde levels and elevating the levels or gene expressions related to specific antioxidative markers. These antioxidation responses elicited by dietary MsYbP influence the hepatic histomorphology and improve liver health of largemouth bass. In conclusion, the implications of our study extend to the aquaculture industry, providing valuable insights into the use of yeast-based paraprobiotics as a nutritional supplement to enhance the antioxidative capacity and liver health of farmed fish.
Minor comments
- L22 & 24: add p value.
Responses: Added as suggested, as line 17-27: The results indicated that MsYbP-fed fish exhibited a significant increase in growth performance in both studies (P < 0.05). Additionally, dietary MsYbP led to a noteworthy reduction in liver function parameters (ALT, AST, and AKP, P < 0.05) and changes in hepatic nuclear density, in-dicating improved liver health. Furthermore, MsYbP inclusion in the diet elevated antioxidative capacity by reducing malondialdehyde levels and increasing levels and gene expressions related to antioxidative markers (e.g., T-AOC, SOD, GSH-Px, CAT, nrf2, and keap1) in both studies (P < 0.05). In terms of hepatic immune responses, the lab-scale study showed an increase in inflammation-related gene expressions (e.g., il-1β and tgf-β1), while the pilot-scale study significantly suppressed expressions of genes related to inflammatory response (e.g., tnfα and il-10) (P < 0.05).
- L57-60: repetition, reorganize the paragraph
Responses: Reorganized the paragraph as suggested, as below:
Line 55-59: Regarding the yeast-derived mannan-oligosaccharides (MOS), it has been widely found to modulate intestinal microbiota composition and inhibit the adherence of pathogenic bacteria, and then benefit host health [15, 16]. Generally, dietary yeast-based paraprobiotics has been shown to have beneficial effects on growth and health in var-ious fish species, including but not limited to….
- L65-67: repetition, reorganize the paragraph
Responses: This paragraph mainly introduces the MsYbP product used in our study, and we believe it should be retained.
- Table 1: analyzed values for all diets are required specially for key nutrients such as crude protein
Responses: Added nutrient information as suggested:
Line 103-110
2 The fishmeal was purchased from Tecnol´ogica de Alimentos S.A., Ltd. (Peru), crude protein 67.0%, crude lipid 10.1%.
3 The cottonseed protein concentrate was purchased from Xinjiang Jinlan Vegetable Protein Co. Ltd. (China), crude protein 65.3%, crude lipid 0.5%.
4 The Clostridium autoethanogenum protein was purchased from Beijing Shoulang Biotechnology Co., Ltd, Beijing (China), crude protein 81.2%, crude lipid 0.3%.
5 The soybean meal was purchased Yihai Kerry Investment Co. Ltd. (China), crude protein 46.6%, crude lipid 0.55%.
- L190-191: rewire.
Responses: Rewired as suggested:
Line 195-197: In the lab-scale study, there was no significant difference in survival rate between Control group and YANG group (Table 3, P > 0.05). However, YANG-fed fish showed significantly higher feed consumption that those in the Control-fed fish (Table 3, P < 0.05).
- Table 3: add P value for example 0.001 not 0.00.
Responses: revised as suggested.
- 2: there is no * used.
Responses: revised as suggested.
